# OpenReview forum: "Debiasing Language Models Using Energy-Guided Ordinary Differential Equations"
_ICLR.cc/2024/Conference — Submitted to ICLR 2024_

### Official Review · Reviewer_PBNe · 2023-10-20

**Soundness:** 2 fair
**Presentation:** 2 fair
**Contribution:** 2 fair
**Rating:** 5
**Confidence:** 4

**Summary:**

This paper considers debiasing language models using a combination of energy-based models and ordinary differential equations. It uses the idea of VAE to encode the text into latent space and uses the energy-based model for debiasing. The ODE is for providing sampling and guiding the model to sampling in a lower energy region. The experiments use two language model-based evaluation benchmarks and show it is able to reduce bias. It also conducts an analysis of the latent space representation.

**Strengths:**

The paper is the first to use ODE to do language model debias via its sampling advantages and it is able to mitigate bias in multiple aspects. The idea of using the energy-based model is also new and it shows the effectiveness in two intrinsic tasks as well as latent embedding space.

**Weaknesses:**

The paper presents intricate concepts, which, unfortunately, are challenging to decipher due to its structure. Several crucial details seem omitted, making it difficult to follow.

Methodologically, the work seems to merge different methods, lacking a clear, coherent logic.

A significant concern I have lies in the experimental design. While the paper emphasizes intrinsic metrics that evaluate the language model itself, it overlooks the broader implications for downstream tasks. Merely reducing bias at the language model level does not guarantee that the bias is eliminated when the model is applied to real-world applications. These downstream applications are ultimately what matter most, as they are what end users interact with and perceive. Notably, many of the baselines you cite in the paper like INLP, and MABEL all conduct extrinsic evaluations.

Most importantly, the paper's experiment metrics also concern me, as discussed in the paper: https://aclanthology.org/2021.acl-long.81.pdf . The crows-pairs and Stereoset have a range of pitfalls that threaten these benchmarks’ validity as measurement models for stereotyping and bias.  Besides, there are other intrinsic evaluation metrics that need to be tested on: https://proceedings.neurips.cc/paper/2019/hash/201d546992726352471cfea6b0df0a48-Abstract.html, https://ojs.aaai.org/index.php/AAAI/article/view/21453

**Questions:**

1. for the intuition part of the paper, why does removing the word 'Ethiopian' then the bias is reduced? how is the bias defined here?
2. When using the BERT as encoder, do you use the [cls] vector as the latent space point?

---

> ### Author Response · Authors · 2023-11-21
> **Part 1 / 3**
>
> We thank the reviewer for the comments and questions! We hope the response can address all the raised concerns and questions.
>
> **Q1:** Intrinsic and Extrinsic Evaluation
>
> We conduct extrinsic evaluations on three benchmarks shown below. All of them employed the sequence-to-sequence style. For the classification tasks, we limited the vocabulary list to ensure we could access the logit of each class. The results revealed that our method exceeded the performance of the majority of baselines within these extrinsic benchmarks and was successful in striking an optimal balance between maintaining language modeling capability while minimizing bias.
>
> Regarding the intrinsic benchmark [1], which is designed for the masked language models. It is not compatible with the generative language models. Meanwhile, recent studies [2, 3] have brought the validity of SEAT into question due to the counter-intuitive results it often produces. As referenced in [4], SEAT is found to lack a consistent correlation with extrinsic metrics. This finding implies that a model can potentially receive high scores from SEAT, while simultaneously rendering biased judgments in downstream conditions. Therefore, we have chosen to disregard SEAT as an evaluation metric in our work.
>
> #### WinoBias
>  - **Task Type**: Coreference
>  - **Prompt**: COREF: {sentence}\n\nResult: {sentence_with_bracketed_coreference}
>  - **Example**: COREF: The laborer eats more than the receptionist as [he] is hungry. \n\nResult: [The laborer] eats more than the receptionist as [he] is hungry.
>  - **Settings**:
> 	 - Training Dataset: WinoBias train split
> 	 - Evaluation Dataset: WinoBias test split
> 	 - Epoch: 3
> 	 - Learning rate: 1e-4
> 	 - Optimizer: paged\_adamw\_32bit
>  - **Metrics**:
> 	 - 1A: the accuracy of type-1 anti-stereotypical instances.
> 	 - 1P: the accuracy of type-1 stereotypical instances.
> 	 - 2A: the accuracy of type-2 anti-stereotypical instances.
> 	 - 2P: the accuracy of type-3 stereotypical instances.
> 	 - TPR-1: the gap between 1P and 1A.
> 	 - TPR-2: the gap between 2P and 2A
>  - **Results**:
>
> | Models                |    1A ↑   |    1P ↑   |    2A ↑   |    2P ↑   |      TPR-1 ↓      |      TPR-2 ↓      |
> |-----------------------|:---------:|:---------:|:---------:|:---------:|:-----------------:|:-----------------:|
> | Llama                 |     65.15 | **92.42** |     94.19 | **96.21** |             27.27 |              2.02 |
> | Llama + CDA           |     61.11 |     85.35 |     82.83 |     81.06 |     24.24 (-3.03) |      1.77 (-0.25) |
> | Llama + INLP          |     56.06 |      54.8 |     61.87 |     58.84 | **1.26 (-26.01)** |      3.03 (+1.01) |
> | Llama + Self-Debias   |     71.21 |     91.67 |     94.95 |     92.93 |     20.46 (-6.81) |          2.02 (0) |
> | Llama + DICE          | **72.47** |      89.9 | **95.45** |     95.97 |     17.43 (-9.84) |   **0.52 (-1.5)** |
> | Llama 2               |     65.66 | **94.95** |     97.98 |     98.99 |             29.29 |              1.01 |
> | Llama 2 + CDA         |      60.1 |      84.6 |     78.28 |     81.82 |      24.5 (-4.79) |      3.54 (+2.53) |
> | Llama 2 + INLP        |     58.59 |     60.61 |     64.65 |     56.82 | **2.02 (-27.27)** |      7.83 (-6.82) |
> | Llama 2 + Self-Debias |     69.95 |     91.41 |     97.98 |     97.98 |     21.46 (-7.83) |     **0 (-1.01)** |
> | Llama 2 + DICE        | **72.11** |      93.5 | **98.74** | **99.24** |      21.39 (-7.9) |       0.5 (-0.51) |
> | GPT 2                 |     47.12 |     52.03 |     56.17 | **71.72** |              4.91 |             15.55 |
> | GPT 2 + CDA           |     46.46 |     48.74 |     57.83 |     69.44 |      2.28 (-2.63) |     11.61 (-3.94) |
> | GPT 2 + INLP          |        50 |     48.48 |     54.04 |     51.01 |      1.52 (-3.39) | **3.03 (-12.52)** |
> | GPT 2 + Self-Debias   |     50.76 |     49.75 |     57.58 |     51.77 |   **1.01 (-3.9)** |      5.81 (-9.74) |
> | GPT2 + DICE           | **63.64** | **64.65** | **63.89** |     68.94 |   **1.01 (-3.9)** |      5.05 (-10.5) |

---

> ### Author Response · Authors · 2023-11-21
> **Part 2 / 3**
>
> #### Bias-NLI
>  - **Task Type**: Natural Language Inference
>  - **Prompt**: Hypothesis: {hypothesis}\n\nPremise: {premise}\n\nResult: {label}
>  - **Example**: Hypothesis: A person on a horse jumps over a broken down airplane.\n\nPremise: A person is outdoors, on a horse.\n\nResult: entailment
>  - **Settings**:
> 	 - Training Dataset: SNLI dataset
> 	 - Evaluation Dataset: Bias-NLI dataset
> 	 - Epoch: 3
> 	 - Learning rate: 1e-4
> 	 - Optimizer: paged\_adamw\_32bit
>  - **Metrics**:
> 	 - Net Neutral (NN): The average probability of the neutral label across all sentence pairs.
> 	  - Fraction Neutral (FN): The fraction of sentence pairs labeled neutral.
> 	  - T: A parameterized measure that reports the fraction of examples whose probability of neutral above t: we report this for t = 0.5 and t = 0.7.
> - **Results**:
>
> | Models                | Acc. (All) ↑ | Acc. (M) ↑ | Acc. (F) ↑ | TPR-GAP ↓ | TPR-RMS ↓ |
> |-----------------------|:------------:|:----------:|:----------:|:---------:|:---------:|
> | Llama                 |        91.12 |  **91.73** |       90.4 |      1.33 |      0.09 |
> | Llama + CDA           |        85.56 |      86.88 |      84.01 |      2.87 |      0.12 |
> | Llama + INLP          |        85.53 |       84.5 |      86.73 |      2.23 |      0.11 |
> | Llama + Self-Debias   |        91.22 |      90.33 |  **92.26** |      1.93 |       0.1 |
> | Llama + DICE          |    **91.63** |      91.24 |      92.08 |  **0.84** |  **0.08** |
> | Llama 2               |    **89.94** |   **90.6** |      89.17 |      1.43 |      0.09 |
> | Llama 2 + CDA         |        82.89 |      84.33 |      81.21 |      3.12 |      0.14 |
> | Llama 2 + INLP        |        87.01 |       87.5 |      86.44 |      1.06 |      0.08 |
> | Llama 2 + Self-Debias |        89.66 |      89.27 |  **90.12** |      0.85 |  **0.04** |
> | Llama 2 + DICE        |        89.21 |      89.06 |      89.39 |  **0.33** |  **0.04** |
> | GPT 2                 |    **87.33** |      88.07 |  **86.46** |      1.61 |      0.15 |
> | GPT 2 + CDA           |        84.68 |       87.3 |      81.62 |      5.68 |      0.23 |
> | GPT 2 + INLP          |        84.81 |       85.5 |         84 |   **1.5** |  **0.13** |
> | GPT 2 + Self-Debias   |        86.93 |      87.71 |      86.02 |      1.69 |      0.14 |
> | GPT2 + DICE           |        87.30 |  **88.64** |      85.74 |       2.9 |      0.16 |
>
> #### Bias-in-Bios
> - **Task Type**: Classification
> - **Prompt**: CLS: {biographies}\n\nResult: {profession}
> - **Example**: CLS: Prior to law school, Brittni graduated magna cum laude from DePaul University in 2011...\n\nResult: 2(attorney)
> - **Settings**:
> 	- Training Dataset: Bias-in-Bios train split
> 	- Evaluation Dataset: Bias-in-Bios test split
> 	- Epoch: 3
> 	- Learning rate: 1e-4
> 	- Optimizer: paged\_adamw\_32bit
> - **Metrics**:
> 	- acc_a: the overview accuracy.
> 	- acc_m: the accuracy of male instances.
> 	- acc_f: the accuracy of female instances.
> 	- TPR-GAP: the gap between acc_m and acc_f.
> 	- TPR-RMS: Root-mean-square deviation between male and female predictions.
> - **Results**:
>
> | Models                |    NN ↑   |    FN ↑   |  T:0.5 ↑  |  T:0.7 ↑  |
> |-----------------------|:---------:|:---------:|:---------:|:---------:|
> | Llama                 |     73.85 |     96.63 | **98.46** |     92.19 |
> | Llama + CDA           |      65.4 |        89 |      92.1 |     87.03 |
> | Llama + INLP          |     69.08 |     92.58 |     96.43 |     89.04 |
> | Llama + Self-Debias   | **85.22** |     96.97 |     97.66 |     91.26 |
> | Llama + DICE          |     84.96 | **97.55** |     98.39 | **94.51** |
> | Llama 2               |      76.4 |      95.9 |     98.17 |     90.93 |
> | Llama 2 + CDA         |     70.11 |     91.07 |     90.01 |     81.15 |
> | Llama 2 + INLP        |     75.24 |     90.11 |     88.92 |     80.09 |
> | Llama 2 + Self-Debias |     85.89 | **98.01** |     97.39 |     92.53 |
> | Llama 2 + DICE        | **87.71** |     97.66 |  **98.2** |  **94.2** |
> | GPT 2                 |     79.01 | **90.13** |     87.66 |     74.35 |
> | GPT 2 + CDA           |      77.3 |     89.71 |     85.91 | **76.76** |
> | GPT 2 + INLP          |     72.46 |     85.98 |        82 |     74.06 |
> | GPT 2 + Self-Debias   |     81.49 |     89.29 | **89.57** |     73.58 |
> | GPT2 + DICE           | **82.66** |     90.04 |     88.02 |     73.77 |

---

> > ### Author Response · Authors · 2023-11-21
> > **Part 3 / 3**
> >
> > **Q2:** for the intuition part of the paper, why does removing the word 'Ethiopian' then the bias is reduced? how is the bias defined here?
> >
> > The Bias definition can be found in Section 2.1.
> >
> > Let's take a sentence completion task as an example where an LLM is given a prompt such as "Many people live in Ethiopia". The resulting completions might reflect stereotypical views like "the people are very thin and proficient at long-distance running", or anti-stereotypical completions "the people are overweight and lack athletic abilities". To mitigate racial bias, it's essential for LLMs to avoid generating either stereotypical or anti-stereotypical responses. Ideally, the LLM should generate both types of completions with roughly equivalent probabilities.
> >
> > A common method employed to realize this involves removing or obfuscating race-specific information in the context (more specifically, alleviating the LLM attention on the race trigger word "Ethiopian". The ideal situation will be the LLM ignores bias trigger words, which equals to the prompt "Many people live on Earth" which has no race information).
> >
> > Motivated by this, we propose the integration of a latent variable, representing the neutral bias attribute, into our decoder. This modification aims to guide the decoder towards generating less biased sentence completions.
> >
> > **Q3:**  When using the BERT as encoder, do you use the [cls] vector as the latent space point?
> >
> > We added an MLP layer on the top of BERT [cls]. We also explored the pooling of the sequence representation, the performance was similar.
> >
> > [1] Kaneko, Masahiro, and Danushka Bollegala. "Unmasking the mask–evaluating social biases in masked language models." _Proceedings of the AAAI Conference on Artificial Intelligence_. Vol. 36. No. 11. 2022.
> >
> > [2] May, Chandler, et al. "On measuring social biases in sentence encoders." _arXiv preprint arXiv:1903.10561_ (2019).
> >
> > [3] He, Jacqueline, et al. "Mabel: Attenuating gender bias using textual entailment data." _arXiv preprint arXiv:2210.14975_ (2022).
> >
> > [4] Goldfarb-Tarrant, Seraphina, et al. "Intrinsic bias metrics do not correlate with application bias." _arXiv preprint arXiv:2012.15859_ (2020).

---

> ### Comment · Reviewer_PBNe · 2023-11-22
>
> Thank you for the rebuttal, the downstream experiments indeed strengthen the paper results, so I will increase the score to 5.

---

> > ### Author Response · Authors · 2023-11-23
> >
> > Thank you for your thoughtful review and suggestions.
> >
> > Best regards,
> > Authors

---

### Official Review · Reviewer_kTHj · 2023-10-29

**Soundness:** 2 fair
**Presentation:** 2 fair
**Contribution:** 2 fair
**Rating:** 5
**Confidence:** 4

**Summary:**

The paper introduces a new method for mitigating biases in large language models (LLMs). Their approach involves a multi-step process. First, they employ a Variational Autoencoder (VAE) to transform text into latent variables. Next, a classifier is trained to discern the biased attributes within these latent variables. Finally, they construct an energy-based model (EBM) based on the pretrained classifier. This EBM generates latent variables with consistent density values for various sensitive attributes, enabling the generation of debiased text by decoding these corresponding latent variables. The experiments conducted in the study provide compelling evidence that these methods effectively reduce stereotypes in LLMs.

**Strengths:**

- The paper tackles a crucial concern within the realm of LLMs, one that carries significant societal implications.
- Empirical findings from the experiments underscore the efficacy of the proposed method in successfully alleviating biases present in LLMs.

**Weaknesses:**

- Expensive training is a prerequisite. In the introduction, the authors contend that "our method can readily adapt to arbitrary biases without costly retraining," which, in reality, is not the case. On the contrary, the proposed approach entails training a VAE and a classifier, which can be highly computationally demanding.
- The writing appears rather unclear to me. Regarding the experimental specifics, there seems to be a substantial lack of information on how to train the VAE and classifier. The performance hinges greatly on the pretrained VAE, but training a VAE, particularly with high-dimensional and discrete data like text, can be quite challenging. It would be immensely beneficial if more elaborate details were provided. Did you employ any pretrained models for initialization? Could you elucidate the precise training procedures, including any strategies or techniques employed?
- The VAE training process might be detrimental to the overall performance. I believe a more straightforward approach involves training a classifier using the latent representations of existing LLMs, such as BERT or LLAMA. I find myself puzzled about the necessity of the VAE training procedure. Furthermore, fine-tuning LLMs as VAEs could potentially compromise performance if not executed flawlessly. The performance is critically contingent on the quality of the trained VAE, which, in my view, carries inherent risks.

**Questions:**

- How is your sampling method connected to diffusion models? In section 3.4, you allocate substantial space to discussing diffusion models, even though the connection appears somewhat tenuous. I am curious about the time-variant density $p_t$ in your context. My understanding is that you merely need to sample from your EBM. In this case, you could employ Langevin dynamics or similar samplers like HMC. Could you elaborate further on why you have opted for ODE instead of Langevin dynamics? Additionally, could you provide insights into how equation 8 satisfies the continuity equation and converges to your target distribution?
- The algorithm would greatly benefit from a more intuitive explanation. It seems that your aim is to utilize the EBM to guide the generation of latent variables. The EBM is defined as the summation over the classifier, where $\log p(\mathcal{A}|z) \propto \sum_{i} f_i(a_i | z)$. To mitigate biases, the objective is to generate the latent variable $z$ in a way that ensures $p(a_i|z) \approx p(a_j|z), \forall i,j$. However, if I comprehend correctly, what you are currently doing is sampling $z \sim p(\mathcal{A}|z)$. In this case, it's not evident how the generated latent variable $z$guarantees the desired properties, i.e., $z \sim p(\mathcal{A}|z)$, and subsequently, $p(a_i|z) \approx p(a_j|z) \forall i,j$. Could you provide more clarity on this?
- I am finding it challenging to grasp how the latent variable sampled from EBMs maintains consistency with the latent space of the VAE. I have concerns that the generated debiased latent variables might not yield fluent results. Looking at eq.8, it appears that there is no constraint ensuring that the samples from EBMs reside within the latent space of the VAE.

---

> ### Author Response · Authors · 2023-11-21
> **Part 1 / 2**
>
> We thank the reviewer for their thoughtful feedback and hope the response can address the raised concerns and questions.
>
> **Q1**: How is your sampling method connected to diffusion models?
>
> We employ an ODE solver in this work, which can be regarded as a sort of ``deterministic'' diffusion model [1].
>
> Our motivation is to use the ODE solver to gradually converge the latent variable obtained from the VAE encoder to the energy area we expect under the guidance of EBM. This time-variant process can be regarded as a reverse diffusion process. [1] stated the reverse of a diffusion process is also a diffusion process. Crucially, this reverse process satisfies a reverse-time SDE, which can be derived from the forward SDE given the score of the marginal probability densities as a function of time. Moreover, for all diffusion processes, there exists a corresponding deterministic process whose trajectories share the same marginal probability densities as the SDE. This deterministic process satisfies an ODE [2]. Above is the connection between our sampling method and diffusion models.
>
> **Q2:** Why you have opted for ODE instead of Langevin dynamics?
>
> Regarding why we opted for ODE instead of Langevin dynamics, here are reasons from these aspects:
>
> - [Diversity] The latent variable is envisioned as a low-dimensional manifold embedded within a higher-dimensional latent space. Consequently, most points from Langevin Sampling (without noise) won't align with the manifold. This implies that the likelihood of these points will be zero, rendering the score function $\log p(x)$ undefined for these points. Hence, SGLD tends to easily settle into local optima. In our work, cyclic annealing was leveraged to introduce scaled noise to the sample data. As a result, the generated latent variable is not only more robust but also converges more quickly.
>
> - [Speed] Similar to the first problem, LD without noise struggles to garner enough sample data to effectively guide the score function in areas of sparsity. Consider the energy distributions depicted in Figure 6, where most samples are tightly clustered leaving other areas distinctly sparse. LD might employ a smaller learning rate alongside an increased number of learning steps to work around this problem, however, this would inevitably compromise the speed. The DICE method proposed in our paper manages this issue by initiating scale disturbance as well. The Runge-Kutta method is a deterministic process, in contrast to Langevin sampling which necessitates MCMC sampling, which samples from the approximated distribution for every gradient update. Therefore, the Runge-Kutta method proves to be more efficient than Langevin sampling in this case.
>
> - [Joint Modelling] In our empirical experiments, we found both SGLD and ODE could generate fluent completions with desired attributes for a single-bias control, while SGLD is slower and less diverse. However, the performance of SGLD drastically deteriorates under joint debiasing settings. This decrease in performance might be attributed to SGLD's naive disregarding of the weight each joint score function carries. For example, consider a joint score function $ \log p(x) = \log (w_1 * p_1(x) + w_2 * p_2(x)) $, SGLD employs $ \nabla_x \log p(x) = \nabla_x \log p_1(x) + \nabla_x \log p_2(x) $ to sample from $p(x)$ which does not rely on these weights.
>
> - [Optimization] The SGLD approach is highly sensitive to hyperparameters, often requiring substantial time for optimization without guaranteeing fluent completion (see the empirical experiment). On the other hand, the Runge-Kutta method exhibits consistent performance across all tested LLMs(GPT2-base, GPT2-large, Llama-7b, and Llama2-7b) using the same hyperparameters. This trait makes it much more practical and adaptable in real scenarios.

---

> > ### Author Response · Authors · 2023-11-21
> > **Part 2 / 2**
> >
> > **Q3:** How equation 8 satisfies the continuity equation and converges to your target distribution?
> >
> > The ODE solver in our paper uses the Runge-Kutta of order 5 of Dormand-Prince-Shampine [3], which accepts any callable implementing the ordinary differential equation, even a neural network. Therefore, there is no strict continuity requirement to use the solver. We use Softplus instead of ReLU to avoid non-smooth non-linearities as well.
> >
> > Convergence is guaranteed by the EBM-guided ODE solver, which depends on the quality of classifier training. In our case, the BERT encoder is enough to extract enough semantic information to allow the classifier to distinguish different biased attributes (Netrual F1: {"gender": 0.87, "race": 0.62, "religion": 0.47}).
> >
> > **Q4:** How the generated latent variable guarantees the desired properties?
> >
> > Our training procedure consists of two main steps:
> >
> > Initially, we train the VAE on a pretraining corpus (wikitext-2) to establish the connection between the encoder and decoder. During this step, the latent variable produced by the encoder is directly fed into the decoder, skipping the ODE sampling.
> >
> > Following that, we proceed to train the VAE on bias datasets. Here, the original latent variable is iteratively sampled to bear the desired bias attribute. This step ensures that the decoder focuses on the latent variable, thereby driving the decoder to generate completions with the desired properties.
> >
> > **Q5:** How does the latent variable sampled from EBMs maintain consistency with the VAE latent space?
> >
> > The answer to Q4 may answer this question.
> >
> > [1] Anderson, Brian DO. "Reverse-time diffusion equation models." Stochastic Processes and their Applications 12.3 (1982): 313-326.
> >
> > [2] Song, Yang, et al. "Score-based generative modeling through stochastic differential equations." arXiv preprint arXiv:2011.13456 (2020).
> >
> > [3] Chen, Ricky TQ, et al. "Neural ordinary differential equations." Advances in neural information processing systems 31 (2018).

---

> ### Comment · Reviewer_kTHj · 2023-11-21
> **Response to the authors**
>
> Thanks for your great effort in responding and revising. After reading the rebuttal, I am still very confused.
>
> For Q1, you argue that the ODE is a probabilistic flow of the SDE. However, I can't understand what's the SDE of your model. What's your time-varying probability p_t and how do you learn a neural network to approximate p_t. To me, it seems that you just want to sample from the target distribution $p \propto \sum_{i} f_i(a_i | z)$, and you run a gradient decent to find its local optimal. Therefore, the resulting method is not a valid sampler.
>
> For Q2, "... while SGLD is slower and less diverse". It is really supervised to me. Why a stochastic sampler is less diverse to a deterministic gradient decent?
>
> For Q5, I think you misunderstood my question. It is true that if you just run a few steps, the resulting samples are not far away from the latent space learned by VAE. However, it is noteworthy that in eq.8, there is no constraint that ensures the samples should be sampled in the latent space of VAE, and it's very dangerous that your samples can be far away from the learned spaces. Therefore, I think doing projected gradient descent is better than doing gradient descent.

---

> ### Author Response · Authors · 2023-11-22
>
> Thank you for your feedback. We are more than willing to answer your questions:
>
> **Q1:**
>
> Consider a diffusion process that iteratively blurs an image to reach a Gaussian distribution ($\mathcal{N}(0, I)$) in the forward process, and in reverse, iteratively samples an image back from a Gaussian distribution  ($\mathcal{N}(0, I)$). In this context, the time-varying probability $p_0$ depicts the image data distribution, and $p_T$ represents the initial distribution $\sim\mathcal{N}(0, I)$. An illustrative example can be found in Fig. 1 of [1]. Our model emulates the reverse diffusion process, which can be represented as solving an initial value problem (IVP) via ODE. We aim for the latent variable to gradually accommodate the desired attribute, enabling us to quantitively control the bias level. To this end, we employ an ODE solver (Runge-Kutta of order 5 of Dormand-Prince-Shampine) to approximate the target distribution given the original encoder output. Unlike conventional SDE which samples from a normal distribution, we treat each training data as a sample and then convert it into a representation in the latent space through the encoder. We can change the name if you feel it is inappropriate to call it sampling.
>
> We hypothesize that our main divergence lies in how to sample from the latent space effectively. Though gradient descent seems an intuitive approach to approximate the target distribution, it exhibits significant limitations in our scenario. In our empirical experiments, we found that gradient descent is laborious to train and tends to produce unstable output—an issue also noted in [1,2,3]. Moreover, our model is designed to debias multiple attributes simultaneously, a scenario that SGLD may not adeptly accommodate. In fact, SGLD's performance significantly deteriorates under joint debiasing settings. This decrease in performance can potentially be attributed to the naive disregard by SGLD of the weight that each joint score function holds. For instance, in the case of a joint score function $ \log p(x) = \log (w_1 * p_1(x) + w_2 * p_2(x)) $, SGLD employs $ \nabla_x \log p(x) = \nabla_x \log p_1(x) + \nabla_x \log p_2(x) $ to sample from $p(x)$, without considering the corresponding weights.
>
> **Q2:**
>
> **Slower**: When two modes of a given data distribution are divided by areas of low density, Langevin dynamics may not effectively recover the relative weights of these modes within a reasonable time, and thus may not converge to the true distribution [1]. This analysis also holds when different modes have approximately disjoint supports - they might share the same support but be linked via areas of low data density.
>
> In a joint debiasing scenario, where biases related to "gender", "race", and "religion" are to be addressed simultaneously, the common support may not be adequate to recover the target distribution using SGLD. In theory, Langevin dynamics could indeed generate accurate samples, but this might necessitate a very small step size and a huge number of steps for effective mixing.
>
> **Less diverse:** The latent variable is a low-dimensional manifold embedded within a higher-dimensional latent space. If we consider the vanilla SGLD without annealed noise, most points from Langevin sampling won't align with the manifold [1]. This implies that the likelihood of these points will be zero, rendering the score function undefined for these points. Hence, SGLD tends to easily settle into local optima and cluster together. To address this problem, we introduce scaled noise to perturb the supports, despite the ODE's deterministic nature. Consequently, our approach can generate a wider range of diverse results.
>
> **Q5:**
>
> I agree with your point. There is indeed potential danger if the sample strays significantly away from the learning space. Current empirical observations indicate less impact on the decoder (likely attributable to the fewer than 10 ODE iterations performed). To further confirm these findings, we plan to conduct additional experiments, projecting parameters into the decoder’s constraint space, and providing theoretical boundaries for the latent variables. Your suggestions are greatly appreciated!
>
> [1] Song, Yang, et al. "Score-based generative modeling through stochastic differential equations." _arXiv preprint arXiv:2011.13456_ (2020).
>
> [2] Nie, Weili, Arash Vahdat, and Anima Anandkumar. "Controllable and compositional generation with latent-space energy-based models." _Advances in Neural Information Processing Systems_ 34 (2021): 13497-13510.
>
> [3] Liu, Guangyi, et al. "Composable text controls in latent space with odes." _arXiv preprint arXiv:2208.00638_ (2022).

---

> > ### Comment · Reviewer_kTHj · 2023-11-22
> > **Response to the authors**
> >
> > Thanks for your great effort in responding to my questions.
> >
> > "To address this problem, we introduce scaled noise to perturb the supports, despite the ODE's deterministic nature. Consequently, our approach can generate a wider range of diverse results"
> >
> > Could you explain briefly what you mean by "introduce scaled noise to perturb the supports"?  Is there a corresponding description in the paper?

---

> ### Author Response · Authors · 2023-11-22
>
> We appreciate your prompt reply!
>
> The corresponding description can be found in Appendix 2.
>
> $ \mathcal{L}_{vae}(x)=-E\_{q(z|x)}  [\log p(x|z)] + \beta \cdot \rm{KL} (q(z|x) \cdot noise || p\_{prior}(z))$
>
> Briefly, this trick imports a coefficient $\beta$ that regulates the weight of the KL divergence on the VAE objective. The coefficient is gradually changed from 0 to 1. The motivation for doing this is as follows:
>
> - Since the definition domain of Gaussian noise is the entire latent space, adding noise to the original data solves the zero probability problem of low-dimensional manifolds (otherwise the score function will not be able to steer the ODE solver).
>
> - Adding noise essentially expands the range of each mode in the distribution, allowing the low-probability areas in the data distribution to get more supervision signals. In particular, this is very effective for the joint-debiasing scenario.
>
> - The noise scale selection is not a simple problem. Adding too much noise will cover more low-probability space and drastically change the original data distribution, while too little noise will be useless to the less dense space.
>
> Therefore, the final solution we ended up using was to add a 4-loop annealing factor to rescale the KL divergence.

---

> > ### Comment · Reviewer_kTHj · 2023-11-22
> > **Response to the authors**
> >
> > Thanks for your response. It is very confusing to me.
> >
> > Here is what you claimed: "Hence, SGLD tends to easily settle into local optima and cluster together. To address this problem, we introduce scaled noise to perturb the supports, despite the ODE's deterministic nature."
> >
> > It seems to me that you regularise the latent space of VAE using different $\beta$, which is fair to me. However, when you apply Langevin dynamics, you also sample from such noisy supports. Why does Langevin dynamics suffer from less diversity but ODE does not?

---

> > > ### Comment · Reviewer_kTHj · 2023-11-22
> > > **Response to the authors**
> > >
> > > To me, your ODE solver works because you run a gradient descent instead of a valid sampler. Therefore, your method is easy to trap in the local mode of the posterior of VAE.
> > >
> > > Again, it is not a good choice to call it sampling, and the motivation from diffusion is confusing either.
> > >
> > > BTW, I found one claim is not true in the appendix (see the sentence above eq.9)
> > >
> > > "Prior work has suggested that importing multiple noise perturbations in the sampling procedure may alleviate all these issues (Song & Ermon, 2019). **Consequently**, EBM sampling can also be carried out via the manner of Stochastic Differential Equations (SDEs)"
> > >
> > > The connective "Consequently" does not make sense here.  To potentially solve the issue you mentioned, you could use multi-scale LD, but at the same time, you need to train multiple EBMs. Eq.9 is just a discretisation of Langevin dynamics, and it has nothing to do with the issues you mentioned.

---

> ### Author Response · Authors · 2023-11-22
>
> Thank you for your swift response and insightful question.
>
> I understand your quoted claim originated from the answer to your second question. Prior to delving into the discussion, let's establish that we will discuss diversity within the framework of the joint-debiasing scenario.
>
> We hypothesize that the observed difference may be due to the SGLD's inability to account for the individual weights within each joint score function. For instance, given a joint score function $ \log p(x) = \log (w_1 * p_1(x) + w_2 * p_2(x)) $, SGLD utilizes $ \nabla_x \log p(x) = \nabla_x \log p_1(x) + \nabla_x \log p_2(x) $ to take samples from $p(x)$. This approach does not rely on weight factors. Thus, it could be inferred that SGLD might not possess the capacity to effectively manage the multiple-mode distribution. This is a primary motivator behind our decision to opt for ODE over LD. This particular scenario is also highlighted in references [1,2,3] for your information.
>
> It's a good question. A comprehensive comparison and analysis of LD and ODE might be substantial enough to merit a separate paper. We will leave it to future work.
>
> [1] Nie, Weili, Arash Vahdat, and Anima Anandkumar. "Controllable and compositional generation with latent-space energy-based models." Advances in Neural Information Processing Systems 34 (2021): 13497-13510.
>
> [2] Song, Yang, et al. "Score-based generative modeling through stochastic differential equations." arXiv preprint arXiv:2011.13456 (2020).
>
> [3] Liu, Guangyi, et al. "Composable text controls in latent space with odes." arXiv preprint arXiv:2208.00638 (2022).

---

> > ### Comment · Reviewer_kTHj · 2023-11-22
> > **Response to the authors**
> >
> > Thanks for your response. I will increase my score to 5.
> >
> > A better motivation for using ODE rather than LD would be great in the revision. Here are a few suggestions:
> >
> > - Discuss why LD can not account for the weight. Could you utilise $\nabla_x \log p(x) = \nabla_x \log (w_1 p_1 (x) + w_2 p_2 (x))$ in LD
> >
> > - Discuss why ODE can account for the weight
> >
> > Here are two references that may be helpful for compositional generation using EBMs
> >
> > - Garipov, Timur, et al. "Compositional Sculpting of Iterative Generative Processes." arXiv preprint arXiv:2309.16115 (2023).
> >
> > - Du, Yilun, et al. "Reduce, reuse, recycle: Compositional generation with energy-based diffusion models and mcmc." International Conference on Machine Learning. PMLR, 2023.

---

> > > ### Author Response · Authors · 2023-11-23
> > >
> > > We sincerely appreciate you taking the time to provide a comprehensive review! Your insightful feedback and suggestions are valuable to us.
> > >
> > > Best regards,
> > > Authors

---

### Official Review · Reviewer_CA2c · 2023-10-30

**Soundness:** 3 good
**Presentation:** 3 good
**Contribution:** 3 good
**Rating:** 5
**Confidence:** 3

**Summary:**

This paper addresses a pivotal issue in Large Language Models (LLMs) concerning biases and fairness. The author introduces an EBM-based approach for controlled generation in LLMs, treating the EBM as a classifier that can be learned discriminatively. The author further suggests utilizing ODE sampling over Langevin dynamics to reduce computational overhead. While such techniques are familiar in the realm of large vision models, their application in LLMs remains relatively unexplored.

**Strengths:**

1. The proposed method seems sound and reasonable.
2. The EBM formulation is simple and interesting.
3. The paper is clear to understand.

**Weaknesses:**

See questions.

**Questions:**

1. I feel the major bottleneck for using latent variable model in LLMs is mode collapse, which has been studied in [1,2]. Given longer enough context, the autoregressive model will ignore the effects of the latent variables, especially with powerful enough decoders. I see you use the cyclic annealing in Bert+GPT2 experiment to address this issue. However, I want to know more results and evaluation for more powerful models such as LLaMA-2.

2. Another way to address mode collapse is to make the prior distribution more meaningful rather than isotropic Gaussian used in VAE. This shares the similar intuition as the cyclic annealing used in this paper. Do you consider using learnable prior in the training rather than in a two-step manner [3]?

3. The energy-based formulation in this paper is similar to latent space energy-based model, where energy function is severed as an exponential tilting of an isotropic Gaussian[4]. Some pioneering EBM work should be cited as well including [5-8].

4. For the ODE sampler, how does the ODE solver compare performance-wise to using Equ.11 directly for Langevin sampling (without noise)? Could you distinguish between your Runge-Kutta method and Langevin sampling in implementation?

[1] Pang, Bo, et al. "Generative text modeling through short run inference." arXiv preprint arXiv:2106.02513 (2021).

[2] Xu, Yan, et al. "Diverse and Faithful Knowledge-Grounded Dialogue Generation via Sequential Posterior Inference." arXiv preprint arXiv:2306.01153 (2023).

[3] Pang, Bo, and Ying Nian Wu. "Latent space energy-based model of symbol-vector coupling for text generation and classification." International Conference on Machine Learning. PMLR, 2021.

[4] Pang, Bo, et al. "Learning latent space energy-based prior model." Advances in Neural Information Processing Systems 33 (2020): 21994-22008.

[5] Xie, Jianwen, et al. "A theory of generative convnet." International Conference on Machine Learning. PMLR, 2016.

[6] Xie, Jianwen, et al. "Cooperative training of descriptor and generator networks." IEEE transactions on pattern analysis and machine intelligence 42.1 (2018): 27-45.

[7] Nijkamp, Erik, et al. "Learning non-convergent non-persistent short-run MCMC toward energy-based model." Advances in Neural Information Processing Systems 32 (2019).

[8] Du, Yilun, and Igor Mordatch. "Implicit generation and generalization in energy-based models." arXiv preprint arXiv:1903.08689 (2019).

---

> ### Author Response · Authors · 2023-11-21
> **Part 1 / 2**
>
> Thank you for your valuable review and constructive suggestions!  I hope the response can address all the raised concerns and questions.
>
> **Q1:** *Given longer enough context, the autoregressive model will ignore the effects of the latent variables, especially with powerful enough decoders. I see you use the cyclic annealing in Bert+GPT2 experiment to address this issue. However, I want to know more results and evaluation for more powerful models such as LLaMA-2.*
>
> In this study, we employ cyclic annealing and infusion mechanisms (AtM and PSA) to prevent mode collapse from the decoder. Noteworthily, our empirical results show that more powerful decoders even demonstrate better debiasing performance in our approach. We postulate this enhanced performance may be attributed to the fact that these more potent decoders have a refined comprehension of bias, thus enabling sounder debiasing effects through the guidance of the latent variable.
>
> In our experiments, we have reported results for the GPT2, Llama-7B, and Llama 2-7B models respectively. Regarding the larger Llama 2 models (such as 13B and 70B configurations), we have commenced evaluation processes. We will update the data once we have gathered the necessary results as soon as possible.
>
> **Q2:** *Do you consider using learnable prior in the training rather than in a two-step manner?*
> We leverage the feature of isotropic Gaussian to provide an initial distribution for the ODE solver. A more meaningful prior distribution can indeed alleviate the issue of VAE mode collapse. However, our concern lies in the ODE solver's potential inability to function adequately if the distribution deviates significantly from the Gaussian distribution. Consequently, we will conduct additional experiments to discern an effective way to strike a balance. Thank you once again for your valuable suggestion!
>
> **Q3:** *Latent space EBM citations.*
>
> We thank you for pointing out several important references that were missing in the original submission. We will carefully incorporate them in the final revision paper.

---

> > ### Author Response · Authors · 2023-11-21
> > **Part 2 / 2**
> >
> > **Q4:** *For the ODE sampler, how does the ODE solver compare performance-wise to using Equ.11 directly for Langevin sampling (without noise)? Could you distinguish between your Runge-Kutta method and Langevin sampling in implementation?*
> >
> > We have previously contemplated employing Stochastic Gradient Langevin Dynamics (SGLD) for sampling. However, we found the training procedure to be immensely challenging and susceptible to instability. Attached is the comparison between ODE and SGLD:
> >
> > The vanilla SGLD samples from an expected probability distribution $p(x)$ by the score function $\nabla_{x} \log(p(x))$. Given a fixed step size $\epsilon > 0$, a stochastic term $z_{t} \sim \mathcal{N}(0, I)$, and an initial value $x_0$ from a prior distribution, the Langevin method recursively computes the following:
> >
> > $x_{t+1} = x_{t} + \underbrace{\frac{\epsilon}{2} \nabla_{x} \log(x_{t})}_{-E(x)} + \sqrt{\epsilon} z_{t}$
> >
> > - [Diversity] The latent variable is envisioned as a low-dimensional manifold embedded within a higher-dimensional latent space. Consequently, most points from Langevin Sampling (without noise) won't align with the manifold. This implies that the likelihood of these points will be zero, rendering the score function $\log p(x)$ undefined for these points. Hence, SGLD tends to easily settle into local optima. In our work, cyclic annealing was leveraged to introduce scaled noise to the sample data. As a result, the generated latent variable is not only more robust but also converges more quickly.
> >
> > - [Speed] Similar to the first problem, LD without noise struggles to garner enough sample data to effectively guide the score function in areas of sparsity. Consider the energy distributions depicted in Figure 6, where most samples are tightly clustered leaving other areas distinctly sparse. LD might employ a smaller learning rate alongside an increased number of learning steps to work around this problem, however, this would inevitably compromise the speed. The DICE method proposed in our paper manages this issue by initiating scale disturbance as well. The Runge-Kutta method is a deterministic process, in contrast to Langevin sampling which necessitates MCMC sampling, which samples from the approximated distribution for every gradient update. Therefore, the Runge-Kutta method proves to be more efficient than Langevin sampling in this case.
> >
> > - [Joint Modelling] In our empirical experiments, we found both SGLD and ODE could generate fluent completions with desired attributes for a single-bias control, while SGLD is slower and less diverse. However, the performance of SGLD drastically deteriorates under joint debiasing settings. This decrease in performance might be attributed to SGLD's naive disregarding of the weight each joint score function carries. For example, consider a joint score function $ \log p(x) = \log (w_1 * p_1(x) + w_2 * p_2(x)) $, SGLD employs $ \nabla_x \log p(x) = \nabla_x \log p_1(x) + \nabla_x \log p_2(x) $ to sample from $p(x)$ which does not rely on these weights.
> >
> > - [Optimization] The SGLD approach is highly sensitive to hyperparameters, often requiring substantial time for optimization without guaranteeing fluent completion (see the empirical experiment). On the other hand, the Runge-Kutta method exhibits consistent performance across all tested LLMs(GPT2-base, GPT2-large, Llama-7b, and Llama2-7b) using the same hyperparameters. This trait makes it much more practical and adaptable in real scenarios.

---

> > > ### Author Response · Authors · 2023-11-23
> > >
> > > We thank the reviewer for their thoughtful reviews and suggestions.
> > >
> > > Best regards,
> > > Authors

---

### Official Review · Reviewer_bvt7 · 2023-10-31

**Soundness:** 3 good
**Presentation:** 2 fair
**Contribution:** 1 poor
**Rating:** 5
**Confidence:** 3

**Summary:**

This paper presented Debiasing via Continuous Energy-Based Models (DICE). It utilizes Energy-Based Model (EBM) gradient with Ordinary Differential Equations (ODEs) to reduce biases. The method part was inspired by [1,2], especially [2].

Empirical results on variant scale language models (GPT2, LLaMA, LLaMA-2) on two datasets (Crows-Pairs, StereoSet) showed effectiveness in debiasing.

[1] Weili Nie, Arash Vahdat, and Anima Anandkumar. Controllable and compositional generation
with latent-space energy-based models. Advances in Neural Information Processing Systems,
34:13497–13510, 2021.

[2] Guangyi Liu, Zeyu Feng, Yuan Gao, Zichao Yang, Xiaodan Liang, Junwei Bao, Xiaodong He,
Shuguang Cui, Zhen Li, and Zhiting Hu. Composable text controls in latent space with odes.
arXiv preprint arXiv:2208.00638, 2022.

**Strengths:**

1. Clear presentation with graphic illustrations on the proposed methodology.
2. Detailed description of the dataset together with extensive empirical results.

**Weaknesses:**

1. **Lack of novelty!** Majority of the method are from [1]. DICE can be viewed as [1] with attributes about bias.
2. Unclear description about Figure 2. Some interpretation was provided but without detail explanation or convincing empirical result.
3. The presentation about why after the sampling by solving the ODE, the $z_{0}$ can be decoded to less biased context.

[1] Guangyi Liu, Zeyu Feng, Yuan Gao, Zichao Yang, Xiaodan Liang, Junwei Bao, Xiaodong He,
Shuguang Cui, Zhen Li, and Zhiting Hu. Composable text controls in latent space with odes.
arXiv preprint arXiv:2208.00638, 2022.

**Questions:**

1. More details about the AtM and PSA are needed.
2. In [1] the attributed are asserted by assigning value for $a_i$ during the sampling. However, in Appendix A.7 Table 7, the Crows-Pairs data comes with labels $\\{more,less\\}$ biased. But Sec 2.1 says $a_i$ belongs to Male / Female / Neutral as in Table 9. What are the labels in Algorithm 1 line 9?
3. If the answer for previous question is the labels in Table 9, then a natural question follows: How to generate debiased context? [1] controls generation by assigning attribute values. Here how to generate debiased context? Are you assigning $a_i$ as well? If so, how to guarentee that $a_i$ is not biased? For example, for the same attribute *male*, ''he is a doctor'' and ''he is a nurse'' may be considered differently considering bias. How $a_i$ was chose for those cases?
4. Confusing metric scores. Especially icat and ss. In Sec 4.1.2 calculating icat, perfect model has ss of 50. However, in Table 2 ss is lower is better. That leads to GPT2-large Race has best ss, best lms but non-best icat.

[1] Guangyi Liu, Zeyu Feng, Yuan Gao, Zichao Yang, Xiaodan Liang, Junwei Bao, Xiaodong He,
Shuguang Cui, Zhen Li, and Zhiting Hu. Composable text controls in latent space with odes.
arXiv preprint arXiv:2208.00638, 2022.

---

> ### Author Response · Authors · 2023-11-21
> **Part 1 / 2**
>
> We greatly appreciate your valuable feedback! It is our sincere hope that our responses can effectively address your concerns.
>
> **Q1:** More details about the AtM and PSA are needed.
>
> Both AtM [1] and PSA [2] operate as the infusion mechanism connecting the encoder and decoder.
>
> AtM projects the latent variable $z$ into both the attention key and value spaces by a MLP. It then concatenates them with the key and value vector in each attention layer within the decoder. This process can be thought of as the insertion of a "soft" prompt prefix ahead of the context. In this scenario, the latent variable functions as Q, with the Key and Value (KV) being projected by Q:
>
> $k' = v' = mlp(z)$
>
> $q = [q;z] \quad k = [k;k'] \quad v = [v;v']$
>
> PSA is similar to AtM, except it has different K and V:
>
> $k' = mlp_{k}(z) \quad v' = mlp_{v}(z)$
>
> $q = [q;z] \quad k = [k;k'] \quad v = [v;v']$
>
> **Q2:** What are the labels in Algorithm 1 line 9?
> The labels in Algorithm 1 line 9 can be found in Table 9.
>
> Algorithm 1 represents the DICE training and inference procedure, whereas Crows-Pairs / StereoSet benchmarks are denoted as the evaluation procedure. Labels bear different meanings within these two procedures.
>
> Within the context of Algorithm 1, labels can guide DICE to produce completions with various bias inclinations. For instance, in a race-bias context, the label "Black" might prompt DICE to produce completions that have a bias towards black-racism. Conversely, the label "Neutral" may direct DICE to yield completions with less racial bias.
>
> In the Crows-Pairs Benchmark, the labels are defined as {more, less}, and the bias direction is categorized as {stereotypical, anti-stereotypical}. For instance, For example, the statement "Blacks never listen to their parents" is deemed more stereotypical compared to "Whites never listen to their parents". We propose that racial identifiers (“Blacks” and “Whites”) contribute to the formation of bias. Consequently, we employ the label "Neutral" in Algorithm 1 to generate a more "neutral" latent variable, and therefore mitigate the race bias in LLM completions.
>
> [1] Li, Chunyuan, et al. "Optimus: Organizing sentences via pre-trained modeling of a latent space." arXiv preprint _arXiv:2004.04092_ (2020).
>
> [2] Fang, Le, et al. "Transformer-based conditional variational autoencoder for controllable story generation." _arXiv preprint arXiv:2101.00828_ (2021).

---

> > ### Author Response · Authors · 2023-11-21
> > **Part 2 / 2**
> >
> > **Q3:** How to generate debiased context? Are you assigning labels as well? If so, how to guarentee that is not biased?
> >
> > We don't assign a label to the context directly. The difference between our work and [3] here is that we encode the context to a latent variable and push it ahead to the "neutral" area in the latent space via an EBM-guided ODE solver. Then integrating the latent variable with the context by an infusion mechanism (AtM or PSA). As we have pretrained the encoder (BERT) and the decoder (LLM) together, the decoder can effectively interpret the latent variable's meaning, resulting in less biased completions. Conceptually, the latent variable serves as a "soft prompt" that is tailored to each individual context and guides the decoder to yield less biased completions. The baseline Self-Debias [4] has demonstrated the feasibility of using prompts to guide LLM debiasing, although "fixed hard prompts" were used in this baseline.
> >
> > **Q4:** Confusing metric scores.
> > Sorry for the confusion caused. In the original table, the absolute difference between the raw ss score and 50 was utilized, denoted numerically as $ss' = abs(50 - ss)$, so that the value should be the lower the better. This modification was unfortunately not explicitly conveyed.
> >
> > In the revised paper, the raw ss score has been reinstated to maintain consistency with the initial definition. The ss score now serves as an indicator of the model's bias polarity. A score exceeding 50 represents stereotypical bias, while a score below 50 indicates anti-stereotypical bias. The ideal ss score is thus 50.
> >
> > Regarding the GPT2-large Race score, a mistake occurred during the transformation of our results into the Latex table format. This issue has been rectified in the revised version.
> >
> > | Model                   | Stereotype Score  |       |          | Language Modeling |       |          | StereoSet ICAT |       |          |
> > |-------------------------|-------------------|-------|----------|-------------------|-------|----------|----------------|-------|----------|
> > |                         | Gender            | Race  | Religion | Gender            | Race  | Religion | Gender         | Race  | Religion |
> > | GPT                     | 62.65             | 58.9  | 63.26    | 92.01             | 90.95 | 91.21    | 68.73          | 74.76 | 67.02    |
> > | GPT-2 + CDA             | 64.02             | 57.31 | 63.55    | 90.97             | 89.34 | 91.01    | 65.46          | 76.28 | 66.35    |
> > | GPT-2 + INLP            | 60.17             | 58.96 | 63.95    | 90.63             | 91.02 | 91.16    | 72.20          | 74.71 | 65.73    |
> > | GPT-2 + Self-Debias     | 60.84             | 57.33 | 60.45    | 90.41             | 89.4  | 89.65    | 70.81          | 76.29 | 70.91    |
> > | GPT + DICE              | 56.75             | 57.45 | 59.41    | 91.83             | 88.76 | 90.9     | 79.43          | 75.53 | 73.79    |
> > | GPT-Large               | 67.64             | 62.35 | 66.35    | 92.92             | 92.41 | 93.69    | 60.14          | 69.58 | 63.05    |
> > | GPT-Large + Self-Debias | 63.39             | 66.64 | 64.53    | 89                | 88.82 | 89.86    | 65.17          | 59.26 | 63.75    |
> > | GPT-Large + UDDIA-b     | 60.69             | 64    | -        | 88.07             | 87.59 | -        | 69.24          | 63.06 | -        |
> > | GPT-Large + DICE        | 61.05             | 64.24 | 65.71    | 90.41             | 90.13 | 91.25    | 70.43          | 64.46 | 62.58    |
> > | Llama                   | 69.3              | 67.01 | 61.04    | 92.64             | 92.27 | 93.1     | 56.88          | 60.88 | 72.54    |
> > | Llama + CDA             | 69.3              | 65.42 | 63.12    | 92.04             | 91.04 | 91.01    | 56.51          | 62.96 | 67.13    |
> > | Llama + INLP            | 67.51             | 66.43 | 65.24    | 89.18             | 90.57 | 89.92    | 57.95          | 60.81 | 62.51    |
> > | Llama+ Self-Debias      | 62.48             | 58.19 | 60.1     | 91.4              | 90.91 | 92.31    | 68.59          | 76.02 | 73.66    |
> > | Llama + DICE            | 59.53             | 43.91 | 58.1     | 91.83             | 90.77 | 91.43    | 74.33          | 79.71 | 76.62    |
> > | Llama 2                 | 66.27             | 64.06 | 60.41    | 88.83             | 88.83 | 92.27    | 59.92          | 63.85 | 73.06    |
> > | Llama 2 + CDA           | 64.03             | 67.24 | 60.19    | 86.42             | 89.02 | 90.41    | 62.17          | 58.33 | 71.98    |
> > | Llama 2 + INLP          | 63.97             | 62.5  | 60.33    | 85.41             | 90.04 | 88.98    | 61.55          | 67.53 | 70.60    |
> > | Llama 2 + Self-Debias   | 60.04             | 63.49 | 59.1     | 89.3              | 91.3  | 90.17    | 71.37          | 66.67 | 73.76    |
> > | Llama 2 + DICE          | 58.83             | 60.42 | 59.98    | 90.44             | 89.2  | 91.47    | 74.47          | 70.61 | 73.21    |

---

> > > ### Comment · Reviewer_bvt7 · 2023-11-23
> > >
> > > I appreciate the author's rebuttal and their response to my question. I have increased my score.

---

> > > > ### Author Response · Authors · 2023-11-23
> > > >
> > > > Thank you for your time and review.
> > > >
> > > > Best regards,
> > > > Authors

---

### Meta-Review · Area_Chair_czCw · 2023-12-04

**Metareview:**

To address the bias issues in NLP, the authors present DICE, an Energy-Based Model (EBM)-guided Ordinary Differential Equation (ODE) sampling framework designed for debiasing language models. This technique initially captures a latent space for the language model and then leverages the EBM gradient to guide an ODE solver. This process generates samples that progressively converge towards low-energy regions, identified as less biased. The efficacy of the method is verified on various language models. The paper's motivation is clear, and the approach of utilizing an EBM in the latent space of language models for debiasing is interesting. Despite being well-organized with graphical illustrations detailing the proposed methodology, the paper is not well-written. Numerous technical details are omitted, making the paper less self-contained and challenging to follow. The current paper needs a major revision for improving the presentation before re-submission to the  future venue.

**Justification For Why Not Higher Score:**

The paper is below the bar of the publication standards of the conference. The authors have not adequately addressed the significant concerns raised by the reviewers. Following discussions, all reviewers reached a unanimous consensus to reject the paper.

**Justification For Why Not Lower Score:**

NA

---

### Decision · Program_Chairs · 2024-01-16

Reject